

# Plant growth promoting bacteria (PGPB)-induced plant adaptations to stresses: an updated review

Awmpuizeli Fanai[1], Beirachhitha Bohia[1], Felicia Lalremruati[1], Nancy Lalhriatpuii[2], Lalrokimi[1], Rosie Lalmuanpuii[3], Prashant Kumar Singh[2] and Zothanpuia[2]

[1] Department of Biotechnology, Mizoram University, Aizawl, Mizoram, India
[2] Department of Biotechnology/Life Sciences, Pachhunga University College, Aizawl, Mizoram, India
[3] Department of Botany, Mizoram University, Aizawl, Mizoram, India

Corresponding author
Zothanpuia, jpahnamte6@gmail.com

## ABSTRACT

Plants and bacteria are co-evolving and interact with one another in a continuous process. This interaction enables the plant to assimilate the nutrients and acquire protection with the help of beneficial bacteria known as plant growth-promoting bacteria (PGPB). These beneficial bacteria naturally produce bioactive compounds that can assist plants' stress tolerance. Moreover, they employ various direct and indirect processes to induce plant growth and protect plants against pathogens. The direct mechanisms involve phytohormone production, phosphate solubilization, zinc solubilization, potassium solubilization, ammonia production, and nitrogen fixation while, the production of siderophores, lytic enzymes, hydrogen cyanide, and antibiotics are included under indirect mechanisms. This property can be exploited to prepare bioformulants for biofertilizers, biopesticides, and biofungicides, which are convenient alternatives for chemical-based products to achieve sustainable agricultural practices. However, the application and importance of PGPB in sustainable agriculture are still debatable despite its immense diversity and plant growth-supporting activities. Moreover, the performance of PGPB varies greatly and is dictated by the environmental factors affecting plant growth and development. This review emphasizes the role of PGPB in plant growth-promoting activities (stress tolerance, production of bioactive compounds and phytohormones) and summarises new formulations and opportunities.

# INTRODUCTION

Plants and bacteria have a continuous and dynamic co-evolutionary relationship where they interact and communicate through the release of bioactive chemical signals. This interaction can be positive or negative (*Wille et al., 2019*). Positive interactions benefit the plants as they assist in obtaining minerals, phytohormones, and other nutrients. These beneficial bacterial species can also act against phytopathogen by releasing several bioactive compounds thereby helping plants to endure several stressful conditions. On the

contrary, harmful interactions are inimical, since pathogens colonize the plant tissues resulting in the death of the host plants (*Dolatabadian, 2021*; *Adedayo et al., 2022*). Therefore, exploiting these beneficial microorganisms can help sustain the plants against stress that hinders their productivity.

The regulations of soil fertility, the nutrient cycle, and the preservation of plant diversity are all significantly influenced by microorganisms as a component of the soil ecosystem. *Zhou et al. (2020)* state that the rhizosphere, a small region surrounding the plant roots serves as a vital zone for significant biological interaction occurring between the plant and microorganisms. It is a productive area where microorganisms like bacteria, actinobacteria, fungi, algae, and protozoa actively battle for nutrition and space to thrive (*Manghwar et al., 2023*). The plant growth promoting microorganism (PGPM) can inhabit and interact with the roots of plants, which is advantageous to both the host and microorganisms; a population of rhizospheric fungi and bacteria has the potential to provide a habitat for other microbes as well (*dos Lopes, Dias-Filho & Gurgel, 2021*). Among all the beneficial microorganisms bacteria are the most abundant, followed by fungi and actinobacteria (*Poria et al., 2021*). They create a positive influence on the plant through nutrient assimilation and acquisition by direct or indirect mechanisms (*Kumari, Meena & Upadhyay, 2018*).

The expansion of the global population puts food security at threat which leads to a rise in the increasing application of inorganic chemical-based fertilizers, detrimental to human health and the environment (*Mitter et al., 2021*), The various environmental stress factors further contribute to the low yielding of crops; therefore, organic farming reliant on microflora like PGPM ensures food availability, enhancing crop productivity, quality, and better environment-friendly agricultural techniques (*da Silva Oliveira et al., 2023*). Hence, crop production utilizing PGPM offers sustainability and safeguards soil biodiversity by minimizing the use of chemical fertilizers (*dos Lopes, Dias-Filho & Gurgel, 2021*).

The different ways of plant growth promotion by bacteria are illustrated in Fig. 1. Among all the bacteria, proteobacteria comprise most plant growth-promoting bacteria (PGPB). It includes genera like *Pantoea, Thiobacillus, Pseudomonas, Micrococcus, Rhodococcus, Azospirillum, Azotobacter, Acinetobacter, Acetobacter Klebsiella, Enterobacter, Alcaligenes, Arthrobacter, Burkholderia, Azorhizobium, Achromobacter, Serratia, Bradyrhizobium, Flavobacterium, Mesorhizobium, Microrhizobium, Streptomyces, Bacillus, Azoarcus, Aeromonas, Azoarcus, Caulobacter, Chromobacterium, Delftia, Frankia, Flavobacterium, Gluconacetobacter, Paenibacillus, Rhizobium* and *Streptomyces dos Lopes, Dias-Filho & Gurgel (2021)*, have all demonstrated that bacteria can boost plant development. Following *Oldroyd et al. (2011)*, the Fabaceae, Poaceae, Asteraceae, Solanaceae, Brassicaceae, and Crassulaceae were the most abundant family connected with the PGPB host plant, they are depicted in Table 1.

The PGPB benefits the plants in several ways, which include aiding several biotic and abiotic stress tolerances, inducing plant growth promotion by solubilizing different inorganic mineral nutrients, nitrogen fixation, the release of plant growth regulators, and several other biochemicals that directly or indirectly favor the plant productivity.

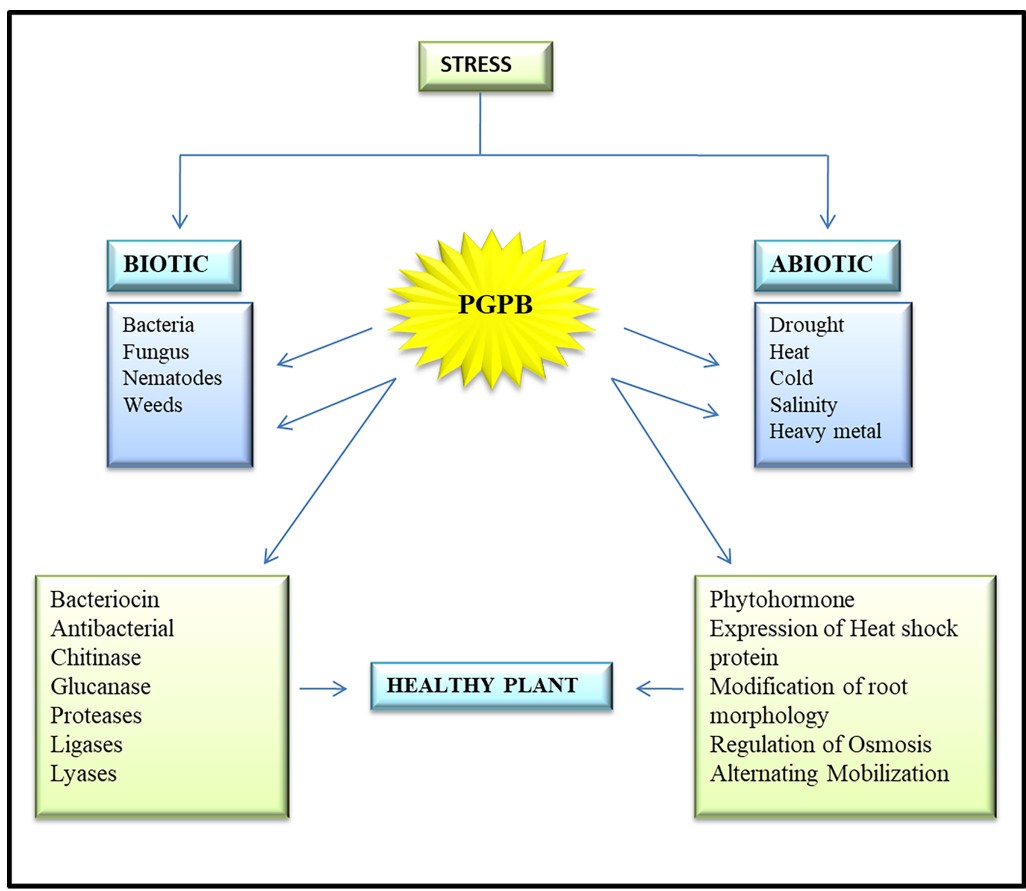

**Figure 1 Various plant growth-promoting activities offered by plant growth-promoting bacteria.** The impact of biotic stress (caused by living organisms) and abiotic stress (environmental factors) on plants highlighting the role of PGBP in mitigating stress effects. PGBP enhance plant growth, nutrient uptake, and stress tolerance through mechanisms such as siderophores, lytic enzyme,HCN, antibiotics & nitrogen fixation, hormone production, *etc.*               

## Search methodology

The main purpose of this review is to emphasize the significance of plant growth-promoting organisms, especially bacteria in the mitigation of various biotic and abiotic factor-induced stress on plants, and their mechanisms employed to improve the quality and quantity of plant products to achieve sustainable agriculture. To ensure the comprehensive and unbiased coverage of the literature, we focus on the latest publications in each of their particular area, between 2000 to 2024 including research articles, review articles, and case studies using Google as a search engine. To elucidate some points, terms like bioinoculants, bioformulants, and bioactive compounds were searched. Some properties of PGPB including bioremediating potential, Pharmaceutical potential, and genetics involved in the plant growth-promoting ability are excluded in this review which may be diverted from this review's goals and confuse the readers.

**Table 1  Plant growth promoting bacteria and their common host plant family.**

| Sl. No | Family | Plant | Plant growth-promoting bacteria | Reference |
|---|---|---|---|---|
| 1 | Fabaceae | *Phaseolus vulgaris* | *Rhizobium acidosoli, R. endophyticum, R. esperanzae, R. etli, R. hidalgoense, R. mesoamericanum, R. tropici, Acinetobacter,* | *Tapia-García et al. (2020)* |
| | | *Mimosa pudica* | *Achromobacter sp., Brevibacillus sp., Cupriavidus sp., Ensifer sp., Stenotrophomonas sp., Pseudomonas sp., Dyella sp., Bacillus sp., Moraxella sp., Rhizobiumj sp.,* | |
| 2 | Poace | Rice, wheat, maize, soirghum, sugarcane | *Azospirillum sp.* | *Pedraza et al. (2020)* |
| 3 | Asteraceae | *Puticaria* | *Bacillus cereus, Agrobacterium fabrum, Brevibacillus brevis, Bacillus subtilis, Paenibacillus, Acinetobacter radioresistant, Burkholderia,* | *ALKahtani et al. (2020)* |
| 4 | Solanaceae | *Artemisia annua* | *Brevibacillus sp., Bacillus sp., Pseudomonas, Azospirillum, Klebsiella, Enterobacter, Alcaligenes, Azotobacter, Streptomyces sp., Pantoea, Bacteroides, Proteobacteria, Radiobacter sp., Stenotrophomonas sp.* | *Husseiny et al. (2021)* |
| 5 | Brassicaceae | *Brasicaoleraceae* | *Pseudomonas sp., Enterobacter., Arthrobacter sp., Pantoea* | *Ferrari et al. (2023)* and *Gustab et al. (2024)* |
| 6 | Crasulaceae | *Echevarilaui* | *Erwinia sp., Pantoea sp.* | *Emmer et al. (2021)* |

### Rationale and intended audience

Agricultural system using beneficial bacteria as bioinoculants is a promising way of achieving a sustainable future. It is a good alternative to chemical fertilizer as an efficient, eco-friendly, productive fertilizer that may aid provide the food demand of the growing global population. They not only enhanced the plant growth but also maintained the soil fertility and enhanced the soil microbiome.

## PLANT GROWTH PROMOTING BACTERIA INDUCED STRESS TOLERANCE

The continual exposure of plants to various stresses (both biotic and abiotic) adversely impacts their growth and development, resulting in compromised yield and quality (*Singh et al., 2021*). As a result, plants develop specific types of defense mechanisms for stress response, which is assisted by a naturally occurring PGPB boosting the resistance against various phytopathogens *via* producing biochemicals and enhancing the soil fertility (*Ramakrishna, Yadav & Li, 2019*; *Leontidou et al., 2020*). The different stress factors and mechanisms of bacterial stress tolerance are depicted in Fig. 2.

### Abiotic stress factor

Climate change induces several abiotic stresses such as drought, salinity, temperature, and heat. These along with nutrient limitations and the presence of heavy metals are crucial players behind compromised yield by crop plants (*Naing, Maung & Kim, 2021*). In the following points, we will discuss the abiotic stress tolerance mechanisms offered by PGPB.

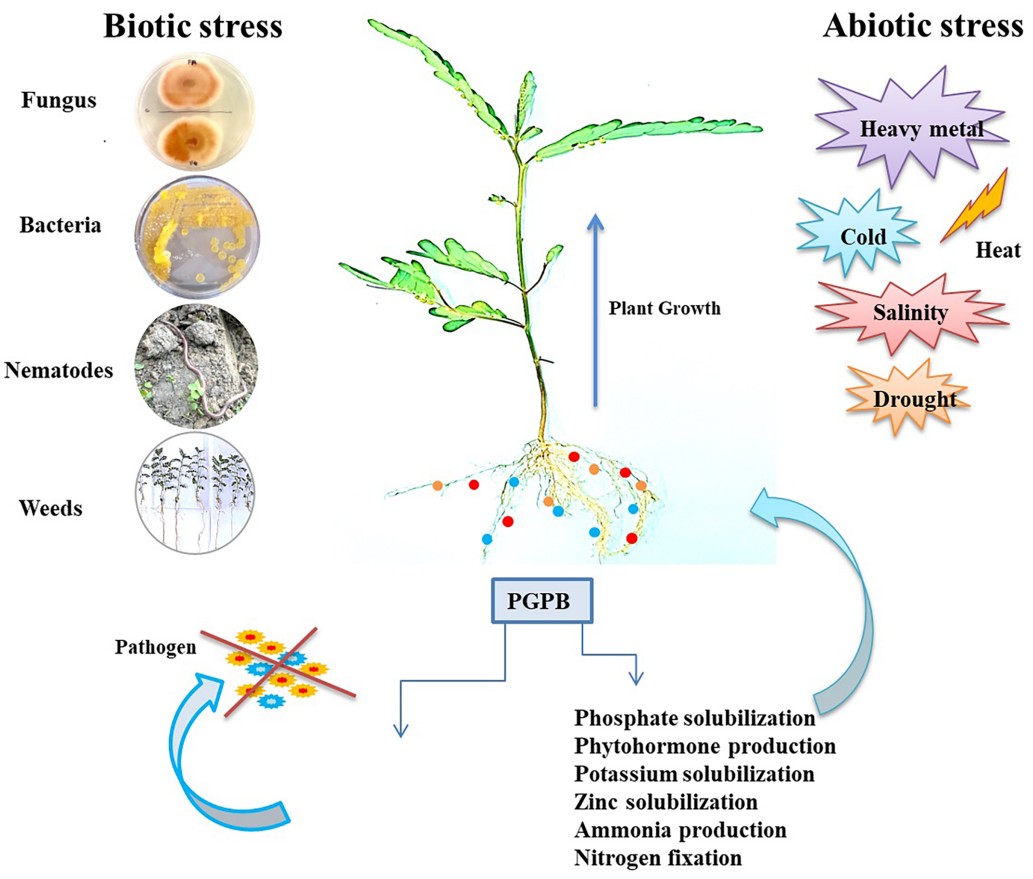

**Figure 2 Different stress factors and response of plant growth promoting bacteria to offer plant growth promotion.** The impact of biotic stress (caused by living organisms) and abiotic stress (environmental factors) on plants highlighting the role of PGBP in mitigating stress effects. PGBP enhance plant growth, nutrient uptake, and stress tolerance through mechanisms such as nitrogen fixation, hormone production, and biocontrol.

## Drought

According to *Ahluwalia, Singh & Bhatia (2021)*, drought stress is divided into four types, a hydrological drought, a socio-economic drought, a meteorological drought and agricultural drought. A hydrological drought arises when there is scarcity and limited water supply in a particular place; a socio-economic drought occurs when water resources are insufficient to meet the demand, meteorological drought happens in dry weather places, and an agriculture drought results due to the reduction of water level in the soil. Drought stress affects the plant by developing reactive oxygen species, which negatively influence the plant structure and mechanism (*Nautiyal et al., 2013*). Therefore, certain mitigation processes like the application of biochar, nanoparticles, film farming, drought resistant plant cultivars, however, provide limited advantages to the agricultural system (*Fadiji et al., 2022*).

PGPB can provide the plant with a better adaptation and tolerance to drought by regulating water absorption, modifying the root structure, and producing phytochemicals (*Khan & Bano, 2019*). Maize plants treated with *Bacillus pumilus* and *Pseudomonas putida*

demonstrated tolerance to drought stress and nutrient limitations (*Kálmán et al., 2024*). Likewise, in another study by *Ilyas et al. (2020)*, bacteria such as *B. subtilis* and *A. brasilense* release a distinct amount of osmolytes that increase the drought tolerance in wheat, enhancing seed and plant growth germination. This plant growth-promoting bacteria also adapts to water stresses by releasing various bioactive compounds like indole-3-acetic acid, salicylic acid, dihydroxybenzoicacid (DHBA), 1-aminocyclopropane-1-carboxylic acid deaminase, and exopolysaccharides and regulate plant growth (*Ahmed et al., 2021*). According to *Danish et al. (2020)*, the soil of water-stress maize was inoculated with the drought tolerating bacteria, and a tremendous improvement in the nutrient uptake by shoot and root elongation and increasing the diameter, dry biomass, and chlorophyll content was observed in the plant. The drought-tolerating bacterial strains, *Pseudomonas lini*, and *Serratia plymuthica* inoculation lowers the drought-induced harm and increases the soil aggregate stability (*Zhang et al., 2020*). Moreover, inoculation of *Bacillus albus*, and *Bacillus cereus* increases the seed vigor index (*Ashry et al., 2022*). All these tolerant strain improves the pigment concentration in the plants increasing the photosynthetic efficiency and the antioxidant properties as well (*Saleem et al., 2021*).

Other drought-tolerating bacteria include *Achromobacter xylosoxidans*, *B. pumilus* (*Castillo-Lorenzo et al., 2018*), *P. aeruginosa*, *L. adecarboxylate*, *E. cloacae*, *P. putida*, *A. xylosoxidans* (*Danish et al., 2020*), *Zobellalle denitrificans*, *Endophyticans*, *P. fluorescens* S3X, *Staphylococcus sciuri* (*Khalilpour, Mozafari & Abbaszadeh-Dahaji, 2021*).

### Temperature stress

A rise in the mean temperature of the climate is one of the most serious abiotic stresses endured by plants (*Desaint et al., 2021*) Bacterial species such as *Bacillus cereus, Serratia liquefacience, Pseudomonas putida, P. fluorescens* (*Mitra et al., 2021*), *Burkholderia phytofirmans, Curvularia proturberata* (*Rana et al., 2021*), *Parabulkholderia phytofirmans* (*Issa et al., 2018*), *Bacillus sp.*, and *Pseudomonas sp.* (*Ahmad et al., 2023*) are identified to have a heat tolerance by balancing plant regulators like cytokinins, ACC deaminase, and antioxidant enzymes which control plant absorption of water and induce the expression of heat shock proteins (*Moumbock et al., 2021*).

A diverse amino acid and its compound have been reported to reduce the harmful effect, and also aid plants in response to heat stress (*Santos et al., 2022*). The application of *B. cereus* increased overall plant biomass, chlorophyll content, and expression of heat shock protein (*Khan et al., 2020b*). A study done by *Park et al. (2017)* reveals that *Enterobacter* SA187 inoculation increases the heat endurance in the wheat and *Arabidopsis* plants, thereby increasing the grain yield, height of the plant, and weight of the seed. Furthermore, *B. cereus* increased the carotenoid, protein, ascorbate peroxidase, chlorophyll content, and superoxide dismutase level in the plant (*Bisht & Mishra, 2020*).

Other bacterial strain like *B. thuringiensis, B. subtilis, P. brassicacearum* (*Ashraf, Bano & Ali, 2019*), *B. velenzensis* (*Abd El-Daim, Bejai & Meijer, 2019*), *B. cereus* (*Khan et al., 2020a*).

## Salinity

The salinity of soil is caused by the scarcity of water (*Reints, Dinar & Crowley, 2020*) and a high concentration of NaCl from the compost fertilizer, which is used for sewage and domestic waste treatment (*Gondek et al., 2020*). Soil salinization also occurs when there is a high concentration of soluble ions like bicarbonate, magnesium, sodium, chloride, sulfate, carbonate, and calcium (*Shahid, Zaman & Heng, 2018*). Excessive $Na^+$ concentration can completely change the soil composition, reduce the fertility of the soil (*Manishankar et al., 2018*) germination rate, and decrease the photosynthetic pigment production by changing the structure of chloroplasts (*Ahmed et al., 2020*), moreover inducing ion toxicity and osmotic imbalance (*Krishnamoorthy et al., 2022*). A study by *Taïbi et al. (2016)* shows that salinity stress decreases the overall plant development, fruit yield, weight, and total number of fruits in a single strawberry plant. *Ansari, Ahmad & Pichtel (2019)* also state that a plant's water movement and stomata conductivity declined due to saline stress. The plant growth-promoting bacteria can counteract the adverse effect of salinity by stimulating the stress response, reducing the ROS production, production of Na-binding exopolysaccharides (*Talebi Atouei, Pourbabaee & Shorafa, 2019*) and also producing phytohormones which promote the growth of root cells, enhancing the water intake (*Subramaniam et al., 2020*). According to *Del Rosario Cappellari et al. (2020)*, a salt-tolerating strain *Bacillus amyloliquifacience* GB03 produces a volatile organic compound that significantly increases the stress mitigation by sixfold to the plant, compared to the one that is not exposed to the compounds. The salt-stress-tolerating bacteria include *Streptomyces* sp. (*Tolba et al., 2019*), *Aneurinibacillus aneurinilyticus*, *Paenibacillus sp.* (*Gupta & Pandey, 2019*), *Pseudomonas azotoformance* (*Liu et al., 2021*). Also, a bacterial strain like *Pseudomonas, Bacillus, Enterobacter, Klebsiella, Agrobacterium, Streptomyces*, and *Ochromobactrum* can tolerate sodium chloride up to 150 g/L (*Zhang et al., 2018*).

## Heavy metal

Heavy metals are inorganic soil pollutants that negatively impact plants. Even though heavy metals are toxic in higher concentrations, they are also an essential source of micronutrients (*Ayangbenro & Babalola, 2017*). Industrial effluent, farm, agrochemical, and domestic waste are anthropological sources of heavy metals in the soil (*Kamran et al., 2020*). Heavy metal enters the plant system, adversely affecting the environment and human beings. Therefore, applying phytohormone-producing bacteria is a sustainable way of removing heavy metals from the soil. The bacteria produce plant hormones that alter the root structure, aiding the plant system to tolerate heavy metal stress (*Ashraf et al., 2017*). *Bacillus* sp. is an efficient cadmium accumulator that lowers the availability of $H_2O_2$, $O^-_2$, and malondialdehyde (MDA) which in turn can trigger heavy metal induced reactive oxygen species stress to the plant (*Zhang et al., 2022*). Although heavy metal are considered harmful for ecological health, they can be tailored and employed for plants over all development. *Bacillus subtilis, Azospirillum brasilens*, and *Pseudomonas fluorescens* integrated with nano zinc increase the concentration of nitrogen, phosphorus, and zinc which in turn enhances the over all productivity of wheat crop (*Jalal et al., 2023c*). Also,

co-inoculation of *Rhizobium tropici* and *Bacillus subtilis* along with nano-zinc foliar spray enhanced the chlorophyll content, zinc concentration and the grain yield (*Jalal et al., 2023a*). Plant regulators like IAA (*Jalmi et al., 2018*) and gibberelins (*Sytar et al., 2019*) enhanced the plants' stress tolerance (*Abdelaal et al., 2021*). Bacteria develop mechanisms like biomolecules and biochemical production by modifying heavy metal mobilization in response to heavy metal.

Heavy metal tolerating bacteria includes *B. subtilis, P. brassicacearum, B. Thuringeansis* (*Ashraf, Bano & Ali, 2019*), *B. cereus* (*Asaf et al., 2017*), *Raoultella ornithinolytica, Brevibacterium, Aspergillus* sp., *Trichoderma* sp., *Pseudomonas flourescens* (*Bhatt et al., 2019*), *Enterobacter* sp. (*Naveed et al., 2020*), *Bacillus* sp. (*Khan et al., 2017*), *B. megaterium* MCR-8 (*Hansda, Kumar & Anshumali, 2017*), *Variovorax* sp., *Bacteroidetes bacterium, P. putida* (*Kamran et al., 2020*), *Alcaligenes faecalis*, and *P. syringae* (*Okpara-Elom et al., 2024*).

## Biotic stresses

PGPM can potentially inhibit biotic stress in plants, induced by pathogenic fungi, bacteria, nematodes, insects, and weeds (*Gupta et al., 2021*), this type of resistance is known as Systemic acquired resistance (SAM), also when the plant growth-promoting bacteria elicit the biotic stress by producing elicitors like volatile organic compounds, microbe associated molecular patterns (MAMPs) in, and bioactive secondary metabolites, it is called induced systemic resistance (ISR) (*Romera et al., 2019*; *Dubey et al., 2020*).

According to *Migunova & Sasanelli (2021)*, the PGPB inhibits pathogenic nematodes directly by releasing different lytic enzymes, antibiotics, and volatile organic compounds indirectly through nitrogen fixation, siderophores, and solubilizing phosphate phytohormones production. Bacterial strains such as *Pasteuria penetrans* (*Mohan et al., 2020*) and *Brevibacillus laterosporus* produce proteases that inhibit the nematode *Heteroderaglycines* (*Abd-Elgawad & Askary, 2018*). *Bacillus megaterium* also produces proteases against *M. graminocula* (*Liang et al., 2019*). Also, *P. aeruginosa, P. cepacia*, and *P. fluorescens* have anthelmintic, antimicrobial, antiviral, cytotoxic, and antitumor properties that can fight against phytopathogens (*Bhavya & Geetha, 2021*) Moreover, bacteria like *B. subtilis*, and *B. pumilus* produce chitinase that combats nematodes like *M. hapla* and *Meloidogyne* sp. (*Kohli et al., 2018*) by causing hydrolysis, disrupting chitin synthesis, producing volatile compounds acting as antifungal (*Xie et al., 2020*). *B. amyloliquefaciens* FZB42 produces bacteriocins inhibiting *M. incognita* (*Liang et al., 2019*). According to *Aeron et al. (2019)*, *Anthrobacter nicotianae* and *Bacillus* sp. release volatile compounds that exhibit activity against *M. graminicola* and *M. incognita*. *Lyseni bacillus, Staphyococcus, Pseudomonas*, and *Enterobacter* also have antibacterial, and antifungal properties (*Mamonokane, Eunice & Mahloro, 2018*).

According to *Oleńska et al. (2020)*, the interactions between plants and various bacteria have an array of effects on plant productivity and soil fertility by producing various chemicals like siderophores, phytohormones, and antibiotics, which inhibit the efficacy of various pathogenic fungi, dissolve phosphate in the soil, and produce indole acetic acid, which promotes plant growth. Moreover, the PGPB must possess inherent rhizospheric

competency which enables it to colonize the rhizosphere in addition to the previously stated properties.

PGPB also reduces or stops the effect of specific pathogens that compete for nutrients and interact with certain beneficial microorganisms and indirectly encourages plant productivity (*Benizri et al., 2021*). PGPB synthesizes antimicrobials like bacteriocins, antibacterial proteins, and enzymes that induce both narrow- and broad-spectrum inhibition of bacteria by altering the structural membrane and damaging the cell wall (*Nazari & Smith, 2020*; *Mak, 2018*) The antagonistic effect of PGPR can occur by producing cell wall hydrolases such as chitinase, glucanase, proteases, and ligases that can destroy the pathogenic cell (*Pérez-Montaño et al., 2014*). Other than directly acting as an anti-pathogenic activity, the antibiotic also triggers the induced systematic resistance (ISR) in plants, suppressing the disease and acting as a biocontrolling agent. The pathogenic bacteria mainly belong to bacterial genera like *Erwinia*, *Pectobacterium*, *Pantoea*, *Agrobacterium*, *Pseudomonas*, *Ralstonia*, *Burkholderia*, *Acidovorax*, *Xanthomonas*, *Clavibacter*, *Streptomyces*, *Xylella*, *Spiroplasma*, and *Phytoplasma*. They cause various diseases, including wilting of leaves, spots, galls, blights, and root rot (*Nabila & Kasiamdari, 2021*). Some examples of the most common disease-causing plant pathogens and the PGP strain that suppresses them are listed in Table 2.

## CLASSIFICATION OF PGPB

### Based on their interactions with plants

Based on interactions, PGPB can be categorized into two types, namely free-living rhizobacteria and symbiotic bacteria. The free-living rhizobacteria are present outside the plant cells, while the symbiotic bacteria, also called endophytes, reside in the intercellular spaces of the plant allowing them direct access to the exchange of metabolites (*Turan et al., 2016*).

Following *Djaya et al. (2019)*, endophytic bacteria are the organisms that colonize the internal tissues of plants at least once in any part of their lifetime. Various endophytic bacteria colonize different plant parts such as leaves, stems, roots, and flowers (*Santoyo et al., 2016*). The density of culturable bacterial cells per gram retrieved from the root is higher than that of stem and leaves, then flowers and fruits (*Amend et al., 2019*). According to *Afzal et al. (2019)*, the diversity of bacteria depends upon the conducive conditions, genetic composition, and physiology of the plant parts they colonize. The relationship between plants and endophytes is considered to be symbiotic and interacts with the root more efficiently. Still, recent studies reveal that the microorganism's mutualism or pathogenicity depends on the genetic composition, environmental factors, and co-colonization of bacteria. Therefore, the term endophytes also includes the pathogen colonizing the plant tissue (*Compant et al., 2021*). The endophytic bacteria can be pre-inoculated in the seeds, enhancing the seed quality, increasing the shelf life, and boosting the plant's endurance to specific stresses (*Zapata-Sarmiento et al., 2020*). Besides, the rhizospheric bacteria can penetrate the plant tissues through the cracks in the roots and cause various tissue injuries to the plant due to the continuous growth of the plant (*Sørensen & Sessitsch, 2007*). Endophytic bacteria can also be used as bio-controlling

**Table 2 Biocontrol of disease-causing phytopathogen by plant growth-promoting bacteria.**

| Phytopathogen | Plant Host | Disease-causing | Resistant PGP strain | Reference |
|---|---|---|---|---|
| Meloidogyne incognita | Rice | Root knot nematode | Trichoderma citrinoviride | Tariq et al. (2020) |
| Xanthomonas oryzae | Rice | Leaf blight | Bacillus subtilin strain GB03 | Faizal Azizi & Lau (2022) |
| Colletotrichum gossipii | Cotton | Ramulosis disease | Bacillus amyloliquefaciens, Bacillus velezensis | Ferro et al. (2020) |
| Colletotrichum sp | Tea | Shoot necrosis | Trichoderma cammeliae | Chakruno, Banik & Sumi (2022) |
| Fusarium oxysporum sp. lycopercici | Tomato | Fusarium wilt | Brevibaccilus brevis | Liu et al. (2022) |
| Botrytis cinera | Beans | Chocolate spot | Trichoderma atroviride | Yones & Kayim (2021) |
| Slerotium cepiiorum | Onion | White rot | Trichoderma atroviride | Rivera-Méndez et al. (2020) |
| Fusarium oxysporum | Cabbage | Wilt | Rhizobactrin | Khafagi, El-Syed & Elwan (2020) |
| Fusarium solani, Macrophomina phaseolina | Soybean | Root rot | Bradyrhizobium | Parveen et al. (2019) |
| Phytophthora capsici | Pepper | Blight and fruit rot | Bacillus licheniformis BL06 | Li et al. (2020) |

agents; they help in plant growth directly and indirectly (Hernández-León et al., 2014), such as the production of antibiotics, cell wall degrading enzymes, and pathogen-resistant volatile compounds. They induce systemic resistance and reduce ethylene production (Santoyo et al., 2016). Furthermore, endophytic actinobacteria also generate secondary metabolites, improving the growth and resilience to various environmental stresses (Girão et al., 2019).

The study performed by Pitiwittayakul, Wongsorn & Tanasupawat (2021) shows the endophytes isolated such as Nguyenibacter vanlangensis, Acidomonas methanolica, Asaia bogorensis, Tanticharoeniaaidae, Burkholderia gladioli, and Bacillus altitudinis from the stem of sugarcane from the Nakhon Ratchasima Province in Thailand effectively inhibit the mycelial growth of F.moniliforme AITO1. These isolates also exhibit plant growth promotion properties through ammonia production, zinc, phosphate solubilization, and biosynthesis of auxin and siderophores. This result highlights that endophytes can be potentially used as PGP and antifungal.

Plant growth-promoting rhizobacteria (PGPR) refers to the bacteria that colonize the rhizospheric region, and aid in plant growth and development by producing beneficial metabolites (Santoyo et al., 2021). The bacteria must have the ability to induce plant growth, suppress or stop the pathogen, and should be invasive (More et al., 2022). PGPR affects the plant by producing and releasing secondary metabolites, which can be employed for crop nutrition and protection, increasing the availability and uptake of different micronutrients from the soil and replacing chemical pesticides (Ramakrishna, Yadav & Li, 2019; Zhao et al., 2021). The rhizospheric microbial diversity is determined by the exudates and metabolites secreted by the roots that provide the optimum environment for microbial growth regarding nutrient availability (Zhao et al., 2021; Vives-Peris et al., 2020).

A study by *Dörr, Moynihan & Mayer (2019)* and *Lee et al. (2022)* reported that the tomato seeds inoculated with a PGPR *Rhodopseudomonas palustris* enhance the overall plant post-harvest quality and increase the nutrient availability in the fruits.

### Based on their cell wall composition

According to *Dörr, Moynihan & Mayer (2019)*, the PGPB can also be classified based on the composition of their cell wall; the bacteria that consist of a thick peptidoglycan wall can retain a Gram dye are called Gram-positive, while the ones that have a thin peptidoglycan wall and cannot keep the Gram's dye are known as Gram-negative.

The Gram-positive PGPB includes *B. alveli*, *B. thuringeansis*, *Clostridium novyi*, *C. limosum*, *Symbiobacterium thermophillum*, *etc.*, Gram-negative PGPB includes *Citrobacter sp.*, *C. freundii*, *C. intermedius*, *C. koseri*, *E. coli*, *Enterobacterderogenes*, *Flavobacterium sp.* (*Rodrigues et al., 2016*), *Azotobacter chroococcum*, *A. insignis*, *A. nigricans*, *A. brasilense*, *Az. salinestris*, and *Az. vinelandii* (*Shelat, Vyas & Jhala, 2017*).

## MECHANISM OF PLANT GROWTH PROMOTING ACTIVITY BY BACTERIA

### Antagonistic activity (indirect mechanism)

#### Siderophores production

Siderophores are the ferric-specific ligands produced by bacteria to combat low iron stress and improve plant growth (*Sayyed et al., 2013*). They are classified as hydroxamates, phenolates, and carboxylates on the basis of their iron-binding component (*Nosrati et al., 2018*). Over 250 types of siderophores have been structurally characterized (*Boukhalfa et al., 2003*). Iron is an essential micronutrient for plant and microorganism growth, metabolism, and survival (*de Souza et al., 2015*). Siderophore-producing bacteria have an iron-regulated protein on their cell surface which transports the ferric iron complex thus, iron becomes available for the metabolic process. Siderophores produced by bacteria are the primary source of iron in events of inefficient iron present in plants (*Perez et al., 2019*).

According to *Loaces, Ferrando & Scavino (2011)*, the rhizobacterial ability to release siderophore has conferred various advantages to endophytic bacteria for colonizing the plant roots and excluding other microorganisms from the same environment. The bacteria that produce siderophores are mostly *Bacillus*, *Chryseobacterium Phyllobacterium* (*Bhatt et al., 2019*), *Pseudomonas sp.* like *Pseudomonas fluorescens*, *P. putida*, *P. aeruginosa*, and *P. aureofaciens*.

The siderophore produced by the bacterial isolates can be tested by the method determined by *Passari et al. (2015)*. A bacterial colony is inoculated in the blue agar plates containing chrome azurol S (CAS) agar medium and incubated at 27 °C for 5 days. The colonies with a yellow-orange halo zone were considered positive for siderophore production.

The siderophore-based drugs and siderophores isolated from microbes can be used efficiently to treat beta-thalassemia and certain anemia, iron overload diseases like hemochromatosis and hemosiderosis, iron poisoning (*Pietrangelo, 2003*), antimalarial, desferrioxamine-B is produced by *Streptomyces piosus* which is active against *P.*

*falcipararum* which causes the depletion of iron. This type of siderophore also inhibits the growth of parasites that cause sleeping sickness in humans (*Nagoba & Vedpathak, 2011*) and cancer treatment (*Petrik et al., 2017*; *Ribeiro & Simões, 2019*).

## Production of lytic enzymes

Plant growth-promoting bacteria serve as defendants of other bacterial pathogens by producing several enzymes, they are elaborated in the following.

### *Protease production*

Proteases produced by microorganisms account for two-thirds of all commercial proteases worldwide (*Younes & Rinaudo, 2015*). The proteases from microbes are desirable since they produce a greater yield and are rapid, space-saving, and cost-effective (*Nisha & Divakaran, 2014*; *Ali et al., 2016*). Proteases can be classified into alkaline, acidic, and neutral. *Bacillus sp.* is the most commonly commercially exploited microbe for protease production. An antifungal metabolite from *B. subtilis* subsp. *natto* purified by *Castaldi et al. (2021)* shows that different proteolytic enzymes such as serine protease, and subtilisin act as an antifungal, showing a high active peak when analyzed with liquid chromatography coupled with tandem mass spectrometry. These show the production of protease as a potential defense system to protect plant aginst a threatening pathogen, which in turn indirectly aids the plant development. Several protease-producing bacteria include *B. clausei, B. licheniformis, B. lentus, A. salinivibrio*, and *Cryptococcus aureus*. Streaming processes purify extracellular alkaline proteases like Subtilisin Carlsberg and Subtilisin Novo to obtain end products (*Kalaiarasi & Sunitha, 2009*).

The production of proteases can be screened using a well plate assay method (*Masi, Gemechu & Tafesse, 2021*). The bacterial strains were inoculated on a 1% Skim milk agar plate. The proteolytic activity was confirmed when a clear halo zone was formed. It was expressed in terms of a millimeter. The PSI for protease activity was calculated using the following.

PSI = (Colony diameter + Clear halo zone diameter)/Colony diameter.

The proteases produced from different bacterial sources can be purified using ion exchange and gel filtration chromatography (*Kanmani et al., 2011*; *Sa et al., 2012*). Other than being an antagonizing agent, alkaline protease is also involved in formulations of ointment, gauze, and non-woven tissues (*Awad et al., 2013*). It also treated lytic enzyme deficiency syndrome (*Gupta & Khare, 2007*; *Palanivel, Ashokkumar & Balagurunathan, 2013*). Moreover, bandages immobilized with elastomers are used for burns, wounds, carbuncles, and furuncles (*Palanivel, Ashokkumar & Balagurunathan, 2013*). Intracellularly-produced proteases have contributed to protein turnover, hydrolysis, hormone regulation, and cell differentiation (*Adrio & Demain, 2014*). Industrial sectors have extensively explored numerous bacterial species for synthesizing products like detergent, food and brewing, silk degumming, denture cleaner, and waste management (*Razzaq et al., 2019*).

### Catalase production

Bacteria with catalase activity are critical for the self-defense and protection of the plant roots against hydrogen peroxide. This type of bacteria indirectly assists plant growth during oxidative stress (*Bumunang & Babalola, 2014*). Bacteria such as *B. marinus, B. insolitus, B. sphaericus, B. pasteurii, B. laterosporus, B. badiu*, and *Staphylococcus aureus* are positive for catalase activity (*Talaiekhozani, 2022*). The bacterial catalase activity can be screened using the tube method described by *Kumar et al. (2012)*. Bacterial colonies incubated for 18–24 h were inoculated in a test tube containing 3% $H_2O_2$ and placed in a dark room to observe the bubble formation. The tubes showing a bubble formation were regarded as bacteria having a catalase activity.

### Amylase production

There are three types of amylase: alpha, beta, and gamma. Among these, alpha-amylase is produced mainly by bacteria, fungi, and actinobacteria. These amylases hydrolyse the cell wall of pathogen thereby guarding the host plant against phytopathogen. The majority of these enzymes are produced by endophytes of medicinal plants and crops (*Ismail et al., 2021*). The bacteria that are known to produce a high amount of alpha-amylase are *Bacillus amyloliquefaciens*, *Bacillus licheniformis*, *Bacillus strearothermophilus*, and *Geobacilus bacterium* (*Far et al., 2020*).

Amylase producers can be observed after spot inoculation of bacterial isolates in the starch Agar medium, incubated at 28 ± 2 °C for 7 days. Iodine solution was splashed onto plates, and after 5–10 min of reaction, a definite halo zone was observed (*Mishra & Behera, 2008*).

The alkaline amylase is an essential constituent of liquid and solid detergents. They are mainly used to remove starch-containing stains (*Niyonzima & More, 2014*).

### Urease production

Some soil bacteria can degrade urea in the form of ammonium and nitrate, which plants can later utilize as a source of nitrogen (*Witte, 2011*). *Brink (2010)* can determine urea hydrolysis. The urea-buffer solution (1% urea at pH 6 with 0.00025% phenol red was added to Stuart's urea broth that contained 5 ml of bacterial cultures. Production of ammonia due to increased pH leads to a change of color. Tubes were incubated for 3 to 5 days at 37 °C and 120 rpm on an orbital shaker. The appearance of red or pink from yellow indicates the breakdown of urea by the bacteria.

The ureolytic bacteria can precipitate calcite by increasing pH and producing carbonate ions. This property is exploited for soil nutrient enrichment, concealment of concrete cracks, and various biomineralization approaches (*Cui et al., 2022*).

## Hydrogen cyanide production

Hydrogen cyanide is a highly toxic volatile compound capable of cellular respiration disruption (*Alemu, 2016*). They inhibit pathogenic fungi, nematodes, insects, and termites (*Sehrawat, Sindhu & Glick, 2022*). The HCN produced by rhizospheric bacteria also acts as a controlling agent for weeds by colonizing the plant roots and hindering their growth. It has no adverse effect on the plant host.

Hydrogen cyanide production can be screened according to a method described by *Lorck (1948)*. According to him, bacteria were streaked in an agar media containing 4.4 g L$^{-1}$ of glycine. The Whatman filter paper was soaked in an alkaline prate solution and put on the lid of the culture plate and then inoculated at 28 °C for 3 days. The color change was observed and considered hydrogen cyanide production.

The HCN-producing bacteria mainly belong to *Pseudomonas* and *Bacillus* species (*Voisard et al., 1989*; *Lieberei, Fock & Biehl, 1996*; *Damodaran et al., 2013*), *Bacillus pumilus*, and *Bacillus subtilis* (*Damodaran et al., 2013*).

# DIRECT MECHANISM

## Phytohormone production

Plant hormones are indispensable chemical messengers that direct the plant's ability to react to the environment (*Gutiérrez-Mañero et al., 2001*; *Vejan et al., 2016*). Both plants and microorganisms can carry out the biosynthesis of plant hormones like cytokinins and auxin. The study carried out by *Daud et al. (2019)*, *Swarnalakshmi et al. (2020)* and *Mekureyaw et al. (2022)*. Certain bacteria like *Paenibacillus polymyxa*, *Rhizobium leguminosarum* and *Pseudomonas fluorescens* are known to be cytokine producers. However, the production of cytokinins is not well studied and investigated due to their diverse compound groups, usually present in minute amounts, making them difficult to identify and quantify.

Moreover, studies regarding the gibberellic acid's production as a plant growth promoter are limited; only a few studies were conducted during the last 20 years; from the latest study performed by *Gutiérrez-Mañero et al. (2001)*, bacterial strains *B. pumilus* and *Bacillus licheniformis* produced four forms of gibberellic acid. Among all the plant hormones, IAA is most commonly investigated and regarded as one of the critical PGB traits for plant growth promotion. It is a heterocyclic compound with a carboxymethyl group that induces leaf formation, embryo development, root initiation and growth, phototropism, geotropism, and fruit development (*Chandra et al., 2019*). A carboxymethyl group, acetic acid, is responsible for all the functions performed by IAA (*Mike-Anosike, Braide & Adeleye, 2018*). Certain studies indicated that some rhizospheric bacteria can produce physiologically active IAA, which inspires root elongation, cell division, and plant growth (*Rehman et al., 2020*). The bacteria that produced phytohormone include *Azospirillum* (*Pedraza et al., 2020*; *Raffi & Charyulu, 2020*), *Arthrobacter* spp. *Bradyrhizobium, Bacillus, Pantoea, Rhanella, Burkholderia, Arthrobacter, Herbaspirillum, Pseudomonas, Enterobacter, Mesorhizobium*, and *Brevundimonas* (*Prasad et al., 2019*). Plants and microbes synthesized IAA through several interrelated pathways. One of them is the dependent pathway. Microbes' IAA varies by physiological parameters like pH, temperature, carbon, and nitrogen sources (*Chandra, Askari & Kumari, 2018*). The bacterial strains like *Bacillus paenibacillus polymyxa, Bacillus subtilis, Camamonas acidovorans, Bacillus megatarium, B. simplex*, and *Enterobacter cancerogenus*, produce IAA, which enable the plant to absorb more nutrients from the soil, resulting in the overall enhancement of growth and development in plants (*Goswami et al., 2013*).

IAA production by rhizobacteria can be analyzed as per the method described by *Ahmed & Hasnain (2010)*. Here, the bacterial isolates were grown in nutrient broth supplemented with 0.5% of L-tryptophan at 27 °C for 3 days. The suspension was then centrifuged at 11,000 rpm for 10 min, and collect the supernatant. Further, 1 ml of the supernatant was added to 2 ml of Salkwoski reagent (1 ml of 0.5 M FeCl$_3$ + +50 ml of 35% perchloric acid) and incubated at room temperature in the dark for 30 min. The formation of a pink color indicated that the bacteria produced IAA. Production of IAA is quantitatively determined by taking OD at 530 nm, and the concentration was expressed in μg/ml. This is an efficient protocol commonly used for qualitative and quantitative estimation of IAA production.

## Phosphate solubilization

According to *Satyaprakash et al. (2017)*, phosphorus is considered the second most essential nutrient for the plant; inadequate phosphorus eventually hinders the growth of the plant. Studies have reported the ability of bacteria to solubilizes the inorganic phosphate compounds like tricalcium phosphate, dicalcium phosphate, hydroxyapatite, as well as rock phosphate into a soluble organic phosphate by releasing organic acids (*Verma et al., 2017*) like citric acid and gluconic acid which chelate the cations of phosphate using the hydroxyl and carboxyl groups present in them (*Youssef, 2014*).

Several studies involving phosphorus-solubilizing bacteria for plant growth improvement report the treatment of maize, wheat, and lettuce seeds with phosphate-solubilizing bacteria (PSB) such as *Pseudomonas putida, Azospirillum lipoferum, Bacillus firmus* and *Bacillus polymyxa*, enhance the solubility of phosphorus in the soil (*Mohamed et al., 2019*). Other species like *P. chlororaphis, Serratia marcescens, B. subtilis, B. megaterium, Arthrobacter aureofaciens, Phyllobacterium myrsinacearum, Rhodococcus erythropolis* (*Kaymak, 2011*), *Burkholderia, flavobacterium, Rhizobium, Erwinia, Acetobacter, Micrococcus, Agrobacterium* and *Achromobacterium* (*Youssef, 2014*) were identified as phosphate solubilizing bacteria.

According to studies conducted by *Lin et al. (2023)* in the potato plant, the phosphate-solubilizing bacteria strain *Bacillus megaterium* activated the gene expression responsible for salinity, drought, and heat stress. Furthermore, the bacteria also trigger different metabolic processes in the plant.

Moreover, the application of *B. subtilis, A. brasilense*, and *P. fluorescens* as a single and combined or coupled with different application rates of phosphorous pentoxide increases the overall sugarcane yields (*Rosa et al., 2022, 2023*).

Screening for phosphate solubilization by bacteria can be done by following the method described by *Kesaulya, Zakaria & Syaiful (2015)*.

## Ammonia production

Ammonia production is a remarkable trait of PGPR for plant growth promotion. When ammonia produced by bacteria is accumulated in the soil, it resulted in alkalinity conditions that repress many phytopathogen. Moreover, ammonia supplies nitrogen to the plant, resulting in root and shoot elongation and biomass growth with increased plant

biomass, eventually enhanced the plant growth indirectly (*Bhattacharyya et al., 2020*; *Gohil et al., 2022*). The bacterial isolates can be screened for ammonia production by a method described by *Bhattacharyya et al. (2020)*. The bacteria incubated for 5 days at 30 °C in a broth containing peptone, NaCl, and yeast extract were centrifuged at 10,000 rpm for 15 min, and subsequently, 0.5 ml Nessler reagent was added. The development of a brown and yellow color indicated ammonia production, and the light absorbance was determined using a Spectrophotometer at 450 nm. The ammonia-producing microbe includes *Pseudomonas putida* (*Ahemad & Khan, 2012*), *Klebsiella* sp. (*Ahemad & Khan, 2010*), *Enterobacter asburiae* (*Wickramasinghe et al., 2021*).

## Nitrogen fixation

Nitrogen is considered the most essential nutrient for plant development since it is required for the overall growth and the production of fruits and seeds (*Mahmud et al., 2020*). Plants cannot directly utilize atmospheric nitrogen; therefore, bacteria assist in nutrient uptake by a symbiotic relationship with plant roots or by non-symbiotic bacteria (*Batista et al., 2018*).

Bacterial genera that form a symbiotic relationship with the plant roots include *Bradyrhizobium, Mesorhizobium, Sinorhizobium, Azhorhizobium, Pararhizobium, Neorhizobium*, and *Pseudomonas* (*Nascimento et al., 2019*). A non-symbiotic bacteria includes *Achromobacter, Herbespirillum, Azoarcus* (*Turan et al., 2016*) *Glucanoacetobacter, Azoarcus, Azotobacter, Azospirillum, Acetobacter, Enterobacter, Burkholderia, Pseudomonas, Cyanobacteria* and *Diazotrophicus* (*Basile & Lepek, 2021*).

A study by *Galindo et al. (2024)* shows that the application of microbial consortia such as *A. subtilis* and *A. brasilense* combined with different nitrogen application rates upregulate the root and shoot development, carbon dioxide uptake, transpiration and leaf chlorophyll index. Also, these microbial consortia inoculated in the seed improved the grain yield and nitrogen accumulation of wheat (*Gaspareto et al., 2023*). Moreover, *Bradyrhizobium* sp. and *Bacillus* sp. co-inoculation improve nodule formation thereby enhancing the nitrogen fixation resulting in the overall plant yield of *Vigna unguiculata* (*Galindo et al., 2022*).

## Zinc solubilization

Zinc serves as an essential co-factor for enzyme activity that is involved in plant growth promotion by the microbes; the siderophore production and zinc ion production are also correlated (*Eshaghi et al., 2019*). The optimum zinc concentration in plants is about 30 to 100 mg/kg; below this level results in deficiency (*Fasim et al., 2002*). According to *Singh et al. (2005)*, zinc deficiency resulted in the slow growth and arising of necrotic marks in the plants. Zinc solubilizing bacteria play an essential role in overcoming inadequate zinc availability. The rhizobacteria mostly solubilizes zinc by producing organic acid metabolites, which lowers the pH of the soil where it is produced and iron chelating enzymes (*Fasim et al., 2002*). The plant enzymes like carbonic anhydrase and superoxide dismutase are bound structurally by the zinc.

The microbes like *P.aeruginosa, Gluconacetobacter diazotrophicus, P. striata, P. fluorescense, Burkholderia cenocepacia, S. liquifaciens, S. marcescens, B. thurigeansis, B. aryabhattai* (*Kamran et al., 2017*), *B. subtilis, Thiobacillus thioxidans*, and *cyanobacteria* (*Hussain et al., 2015*) are reported to solubilize zinc. Also, genera like *Rhizobium, Pseudomonas*, and *Bacillus* promote the zinc translocation towards plants from soil, thereby improving the grain yield and zinc biofortification (*Jalal et al., 2022*) These zinc-solubilizing microbes improve the overall quality and productivity of wheat by producing exopolysaccharides and siderophores (*Jalal, Júnior & Teixeira Filho, 2024*). Moreover, co-inoculation of *B.subtilis* and foliar zinc oxide, *P. fluorescence* and foliar zinc oxide improved the the chlorophyll content, an amino acids, grains glutelin and prolamin in maize (*Jalal et al., 2023b*).

The zinc solubilization potential of bacteria can be screened using a modified Pikovskaya Agar containing insoluble zinc oxide. On the medium, 5 μL of bacterial culture was inoculated, then incubated at $28 \pm 2\,^{\circ}\text{C}$; the result can be observed after the 2, 4, and 7 days. The potential for zinc solubilization of the isolate was indicated by developing a distinct halo zone around the bacterial growth spot (*Sharma et al., 2012*). The zinc solubilizing index can be calculated by the ratio of Halo zone formed + colony/colony diameters (*Saravanan, Madhaiyan & Thangaraju, 2007*).

According to a study by *Wu et al. (2013)*, zinc inhibited biofilm production by *A. pleuropneumoniae, Salmonella typhymurium, E.coli, S.aureus*, and *Streptococcus suis* efficiently. The zinc nanoparticles obtained from the bacteria can also be used to enhance the antimicrobial and biocidal activity of the human oral microbiome (*Lallo da Silva et al., 2019*).

## Potassium solubilization

Potassium is naturally present in the soil but they are not readily absorbed by it since it exists in an insoluble form, therefore plant growth-promoting bacteria solubilize potassium by proffering a variety of organic acids including citric acid, oxalic acid, and tartaric acid (*Olaniyan et al., 2022*). Potassium has several critical functions, as it can alter enzymes physical structures and expose the active site for the reactions. Furthermore, it activates at least 60 enzymes involved in plant growth (*Prajapati & Modi, 2012*). According to *Rawat, Pandey & Saxena (2022)*, plants depend on potassium to open and close stomata. The high level of potassium content in plants resulted in improved disease resistance, fiber quality in cotton, and durability of fruit and vegetables and their physical quality (*Prajapati & Modi, 2012*).

Potassium concentration in the plants regulates water retained in the plant, and low potassium content results in sensitivity to water stress. Bacteria including *Acidithiobacillus, Burkholderia* and *Pseudomonas* (*Sharma, Shankhdhar & Shankhdhar, 2016*), *Bacillus megaterium, Arthrobacter* (*Keshavarz Zarjani et al., 2013*), *Pantoea ananatis,Rahnella aquatilis, Enterobacter* sp. (*Bakhshandeh, Pirdashti & Lendeh, 2017*), *Bacillus mucilaginosus, Paenibacillus mucilaginosus* (*Hu, Chen & Guo, 2006*), *Bacillus licheniformis, Pseudomonas azotoformans* (*Maurya et al., 2016*), *Bacillus edaphicus* (*Sheng,*

*Huang & YongXian, 2000*), and *Pseudomonas putida* (*Bagyalakshmi, Ponmurugan & Marimuthu, 2012*) have been identified as a potent K solubilizer.

The bacteria's ability to saturate the potassium can be studied by spot inoculation in Aleksandrow agar medium, the halo zone formed around the bacteria after seven days suggests the potential to solubilize potassium (*Sood et al., 2023*). This protocol is time-friendly, and commonly used for quality testing.

## BIOFORMULATIONS OF PLANT GROWTH PROMOTING BACTERIA

PGPRs significantly interest the agro-industrial sector. They have been utilized and produced primarily for global crops (*Tabassum et al., 2017*). The plant growth-promoting bacteria are formulated to increase their survival rate while stored and applied to the plants. They are developed into liquid or solid-based wet and dry products (*Berger et al., 2018*). Liquid formulation is regarded as the most effective among them, and it is further divided into seed inoculation, soil inoculation, and shoot inoculation (*Lopes, Dias-Filho & Gurgel, 2021*). According to *Lee et al. (2022)*, liquid inoculants are a mixture of a whole culture and a compound like oil, water, and other polymeric compounds that can enhance the stability, adhesion as well as capacity of dispersion. According to *Nosheen, Ajmal & Song (2021)*, these inoculants inhabit the soil environment and the interior part of the plant tissues, enhancing growth and development. In addition to the aforementioned inoculants, *Kaur & Kaur (2018)* elaborate different types of bioformulants, both carrier-based and encapsulated. Carrier-based inoculants employ, clay, sawdust, straw, charcoal, and other biodegradable materials as carriers which can aid in the survival of inoculants. Also, the encapsulated bioformulation recruits a natural encapsulator like agar, agarose, cellulose, biochar, and synthetic encapsulator including polyvinylpyrrolidone (PVP), polystyrene, and polyacrylamides to extend the shelf life of inoculants. A good inoculant must have a long shelf life with high endurability in harsh environments and be compatible with other agrochemical products, in addition to that, they must possess the ability to be introduced to the plant *via* foliar spray, seed treatment, soil application, bio priming, and seed dip (*Ahmad et al., 2022*). These inoculants are biofertilizers and can be classified based on their function.

**i) Nitrogen fixer:** According to *Nosheen, Ajmal & Song (2021)* bacterial species like *Klebsiella, Desulfovbrio, Anabaena, Rhodospirillum, Rhizobium, Frankia, Aulosira bejerinkic, sligonema, Nostoc, Trichodesmium, Acetobacterdiazotrophicus, Clostridium, Azospirillum* spp, *Alkaligenes, Azoarcus* spp, and *Enterobacter* are commonly formulated as a nitrogen-fixing biofertilizer. Among PGPB, *Azospirillum* is an industrially notable microbe developed as a biofertilizer with strong efficacy (*Etesami & Emami, 2017*; *Raffi & Charyulu, 2020*).

**ii) Potassium solubilizers:** Even though there is a significant number of potential potassium solubilizing bacteria, only a few are on the market due to failed survival during the different stages of formulations. According to *Etesami & Emami (2017)* bacteria like *Mucilaginosus, B.circulanscan, Arthrobacter* spp, *B.edaphicus*, and *Bacillus* are being formulated as potassium solubilizers.

**iii) Plant growth promoter:** The bacteria like *Agrobacterium, Eewinia, Pseudomonas fluorescens, Xanthomonas, Enterobacter, Rhizobium, Streptomyces, Arthrobacter* and *Pseudomonas* sp. (*Nosheen, Ajmal & Song, 2021*) are being effectively used.

Biopesticides are another formulation of beneficial bacteria with targeted activity against pathogens, contrary to a chemical pesticide that is non-targeted and causes significant harm to other beneficial organisms. The Environmental Protection Agency (EPA) highly promotes biopesticides since they are environmentally harmless. One of the most commonly exploited bacteria for biopesticide production is *Bacillus thuringiensis* (*DeJong, 2020*).

However, formulation of efficacious bioinoculants is a great challenge, since many of the potential candidates failed to thrive within the formulation as well as after application in the agriculture field. Therefore, Cell-free supernatant (CFSs) with a secondary metabolite is also a desirable biostimulant and biofertilizer for the achievement of sustainable agriculture. According to *Pellegrini et al. (2020)*, *Bacillus* sp. is a capable genus for CFSs. Moreover, *Azospirillum brasilense* (*Berbel et al., 2020*), *Bradyrhizobium diazoefficiens, R. tropici* CIAT889 (*Gustavo Moretti et al., 2019*), *Bradyrhizobium* sp IC-4059 (*Tewari, Pooniya & Sharma, 2020*), and *Lactobacillus rahmnosus* (*Caballero et al., 2020*), are all known for their effective biostimulation. They offer a high potential to act against phytopathogen as they possess a variety of bioactive molecules including, surfactin, subtilin, subtilisin, mycosubtilin, and rhizoctocins (*Pellegrini et al., 2020*).

## CONCLUSION

In today's world of agriculture system, the culturable land is being contaminated and eroded, however, demands for crops increase to feed the growing population, therefore, adoption of sustainable farming is highly essential. Plant growth-promoting microorganisms, specifically bacteria, have been extensively studied for their potential ability to produce essential metabolites that directly or indirectly assist plant growth and development. Furthermore, the PGPM can naturally release essential biochemicals like plant growth regulators, siderophores, hydrogen cyanide, lytic enzymes, ammonia, *etc*., and can solubilize inorganic phosphate, zinc, and potassium. They also have the potential to fix the atmospheric nitrogen and convert it into soluble form using enzymes called nitrogenase. Therefore, these beneficial bacteria are essential in attaining a sustainable agriculture system, enabling us to obtain a high quality and high quantity of plant products in a much safer and environmentally friendly way. However, limited reports have been made on the formulation of these beneficial bacteria for biofertilizers and biocontrolling agents to enhance crop yield under different biotic and abiotic stress.

Plant growth-promoting bacteria hold the most promising way of sustainable agriculture, accordingly further research and exploration of potential plant growth-promoting bacteria for specific stresses and plants, also having a broad spectrum activity can be investigated to extend the productivity of desired crops. Moreover, study on the compatibility of PGPB and the host plants needed more attention to aid eliminating the possible loss of active plant growth promoting traits of PGPB while adapting into the new host plant environment. This can also help in selection of a right microbial strains for a

particular crop of a particular environment. In addition to that, further study on the co-existing potential of different PGPBs can be done, a strain which can coinhabit the same environment without lowering each of their potential active traits to produce more effective microbial consortia.

## ACKNOWLEDGEMENTS

Authors are thankful to Dr. Laldinpuii, Assistant Professor, Department of English, Pachhunga University College, Aizawl, Mizoram for her meticulous proofreading and invaluable English corrections significantly enhanced the quality of this article.

### Funding

This work was supported by the Department of Science and Technology, Science and Engineering Research Board (DST-SERB), Government of India, vide project sanction no.: EEQ/2022/000878; Indian Council of Medical Research (ICMR) under sanction no.: ECD/NER/5/2022-23; UGC SRG F.30-555/202t(BSR) and RPG (11/1-349/2022/FIN-B/). The funders had no role in study design, data collection and analysis, decision to publish, or preparation of the manuscript.

### Grant Disclosures

The following grant information was disclosed by the authors:
Department of Science and Technology, Science and Engineering Research Board (DST-SERB): EEQ/2022/000878.
Indian Council of Medical Research (ICMR): ECD/NER/5/2022-23 and UGC SRG F.30-555/202t(BSR).
RPG: 11/1-349/2022/FIN-B/.

### Competing Interests

Zothan Puia is an Academic Editor for PeerJ.

### Author Contributions

- Awmpuizeli Fanai conceived and designed the experiments, performed the experiments, analyzed the data, prepared figures and/or tables, authored or reviewed drafts of the article, and approved the final draft.
- Beirachhitha Bohia analyzed the data, prepared figures and/or tables, authored or reviewed drafts of the article, and approved the final draft.
- Felicia Lalremruati analyzed the data, authored or reviewed drafts of the article, and approved the final draft.
- Nancy Lalhriatpuii analyzed the data, authored or reviewed drafts of the article, and approved the final draft.
- Lalrokimi analyzed the data, authored or reviewed drafts of the article, and approved the final draft.

- Rosie Lalmuanpuii analyzed the data, authored or reviewed drafts of the article, and approved the final draft.
- Prashant Kumar Singh analyzed the data, authored or reviewed drafts of the article, and approved the final draft.
- Zothanpuia conceived and designed the experiments, performed the experiments, analyzed the data, prepared figures and/or tables, authored or reviewed drafts of the article, and approved the final draft.

## Data Availability

This is a literature review; there is no raw data.

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
