# Peer review of "Plant growth promoting bacteria (PGPB)-induced plant adaptations to stresses: an updated review"

_PeerJ, doi:10.7717/peerj.17882_

## Round 0.1 · original submission · Major Revisions

Dear Dr. Puia,

We have two independent professionals assess your work. They both agreed that it needs to be substantially altered before it can be published in PeerJ. Please read over the reviewers' thorough remarks and respond to each one.

With best regards,

Reviewer 1 ·

Basic reporting

The review “Plant growth promoting bacteria and PGPB-Induced plant adaptations to stresses: an updated review” addresses a topic of great relevance and interest and focuses on the results reported in the literature in recent years.
The manuscript reads smoothly, however it has some problems.
Some paragraphs go beyond the topic covered and need to be completely rewritten. In paragraph 4.1, the production of siderophores for example, lines 375-379 should be deleted. Or the paragraph on the production of proteases: it does not deal at all with the function that these enzymes can have in association with the plant, but deals with topics that are completely off topic. Even paragraph 4.2.3 on the function of amylases does not touch on the function that they can perform in favor of the plant.

Experimental design

No comment

Validity of the findings

When abbreviating the names of bacteria, if they are different species, you need to use a different abbreviation. Ex: Azospirillum brasilense, Arthrobacter globiformis, Azotobacter vinelandii. They can be abbreviated, for example, to A. brasilense, Ar. globiformis and Az. vinelandii.
The adverb “like” is often used. In many cases, when indicating an identity (when you want to specify something) it is better to use “such as”.

In particular:
Line 211. Change “can deteriorate” with “can counteract”
Line 232. “Heavy metal induced Reactive” lowercase “heavy” and “reactive”
Line 248. Same thing
Line 342. Pseudomonadaceae are Gram-negative.
Line 346. Azospirillum brasilense.
Line 358. Plants do not produce iron. They absorb it, they take it up.
Line 457. Which previous study?
Line 459. Plant hormones are cytokinins, not cytokines.
Line 462. GA's. When first cited it should be read in full and not abbreviated.
Lines 472-474. Also include Azospirillum, it is among the most studied bacteria in this regard.
Lines 526-527. The test you propose as a screening for ammonia production does not involve nitrogen fixation because you provide peptone, not N2
Line 561. “On the media,” change to “On the medium”
Line 593. “Plant growth-promoting bacteria are notable in the agriculture industry” better “PGPRs significantly interest the agro-industrial sector.”
Line 594. Change “synthesized” with “produced”.

·

Basic reporting

The article needs improvement.

Experimental design

No comment

Validity of the findings

Needs improvement.

Additional comments

Fanai et al. tried to show PGPBs-induced adaptation to stresses.
I have the following before publication.

1. The title is not appropriate, when the author mentioned plant growth-promoting bacteria then what does PGPB mean? There is duplication.
2. The reference style is not uniform throughout the article.
3. There are some incomplete sentences, please go through the entire article and correct them.
4. Replace the Raaijmakers et al 2009 with “Beneficial Microorganisms Improve Agricultural Sustainability under Climatic Extremes”.

5. L 86-88: The citation is not in the correct place.
6. L 88-91: Please cite the following sentence with “Regulatory mechanisms of plant growth-promoting rhizobacteria and plant nutrition against abiotic stresses in Brassicaceae family”.
7. L 148: revise the sentence.
8. There is lower- and upper-case letters where not needed. Please address this through the article.
L 233-234: This information is unnecessary at this point. There should be some examples of heavy/essential metals, which are improved by these kinds of strains. i.e. Yield, zinc efficiencies, and biofortification of wheat with zinc sulfate application in soil and foliar nanozinc fertilization; Integrated use of plant growth-promoting bacteria and nano-zinc foliar spray is a sustainable approach for wheat biofortification, yield, and zinc use efficiency; Nano-zinc and plant growth-promoting bacteria is a sustainable alternative for improving productivity and agronomic biofortification of common bean. I found these articles to be the best example for your review as the highest zinc doses affect plant growth and yield negatively.
9. The article is extended but the figures and tables are not so much supportive. I will suggest adding another figure regarding the mechanisms.
10. The Phosphate Solubilization section has some old references, I would suggest having a look of the most relevant and recent citations as follows: Inoculation with plant growth-promoting bacteria to reduce phosphate fertilization requirement and enhance technological quality and yield of sugarcane; Technological Quality of Sugarcane Inoculated with Plant-Growth-Promoting Bacteria and Residual Effect of Phosphorus Rates.
11. L: 513-19: Delete this information.
12. Nitrogen has not been discussed in detail however, it’s one of the most extensively studied section with PGPB. Lots of recent literature is available on this important interaction such as, Impact of nitrogen fertilizer sustainability on corn crop yield: the role of beneficial microbial inoculation interactions; Inoculation with Plant Growth-Promoting Bacteria and Nitrogen Doses Improves Wheat Productivity and Nitrogen Use Efficiency; Co-Inoculation with Azospirillum brasilense and Bradyrhizobium sp. Enhances Nitrogen Uptake and Yield in Field-Grown Cowpea and Did Not Change N …; Improving Sustainable Field-Grown Wheat Production With Azospirillum brasilense Under Tropical Conditions: A Potential Tool for Improving Nitrogen Management. All these articles will help you to discuss the role of PGPBs in the reduction of N fertilization for a safe environment.
13. L 550: Please insert the proper year of citation.
14. Zinc has been discussed for around one paragraph, which shouldn’t be the focus of the study instead discuss the role of PGPB in Zn solubilization or the interaction of zinc and PGPBs. Also, the citations provided in the survey have not been updated much. Look into the following recently published reviews and articles: Interaction of Zinc Mineral Nutrition and Plant Growth-Promoting Bacteria in Tropical Agricultural Systems: A Review; Nanozinc and plant growth-promoting bacteria improve biochemical and metabolic attributes of maize in tropical Cerrado; Integrated use of plant growth-promoting bacteria and nano-zinc foliar spray is a sustainable approach for wheat biofortification, yield, and zinc use efficiency; Diazotrophic bacteria is an alternative strategy for increasing grain biofortification, yield and zinc use efficiency of maize etc..
15. I think bioformulation and onward should be the repetition of the above literature. Delete it.
16. Conclusions should be more comprehensive with a proper future direction. The further study that you indicated here has already been done and is available in the literature. Come up with some new ideas to pitch your review.

---

## Round 0.2 · accepted · Accept

Dear Dr. Puia,

The current version of Your manuscript was assesed by independent expert. The reviewer had no comments on the current version, so the paper can be published. Congratulations!

·

Basic reporting

The authors addressed my suggestions and comments satisfactorily.

Experimental design

Not applicable

Validity of the findings

Yes.